# Multiple Ecological Niche Modeling Reveals Niche Conservatism and Divergence in East Asian Yew (*Taxus*)

**DOI:** 10.3390/plants14071094

**Published:** 2025-04-01

**Authors:** Chuncheng Wang, Minqiu Wang, Shanshan Zhu, Xingtong Wu, Shaolong Yang, Yadan Yan, Yafeng Wen

**Affiliations:** 1College of Landscape and Architecture, Central South University of Forestry and Technology, Changsha 410004, Chinamerakyangll@163.com (S.Y.);; 2Hunan Big Data Engineering Technology Research Center of Natural Protected Areas Landscape Resources, Changsha 410004, China; 3Yuelushan Laboratory Carbon Sinks Forests Variety Innovation Center, Changsha 410004, China

**Keywords:** biogeographic history, conservation priorities, niche evolution, phyloclimatic modeling, phylogenetic niche conservatism, *Taxus* in East Asia

## Abstract

Understanding ecological niche evolution patterns is crucial for elucidating biogeographic history and guiding biodiversity conservation. *Taxus* is a Tertiary relict gymnosperm with 11 lineages mainly distributed across East Asia, spanning from tropical to subarctic regions. However, the spatiotemporal dynamics of its ecological niche evolution and the roles of ecological and geographical factors in lineage diversification, remain unclear. Using occurrence records, environmental data, and reconstructed phylogenies, we employed ensemble ecological niche models (eENMs), environmental principle components analysis (PCA-env), and phyloclimatic modeling to analyze niche similarity and evolution among 11 *Taxus* lineages. Based on reconstructed Bayesian trees and geographical distribution characteristics, we classified the eleven lineages into four clades: Northern (*T. cuspidata*), Central (*T. chinensis*, *T. qinlingensis*, and the Emei type), Western (*T. wallichiana*, *T. florinii*, and *T. contorta*), and Southern (*T. calcicola*, *T. phytonii*, *T. mairei*, and the Huangshan type). Orogenic activities and climate changes in the Tibetan Plateau since the Late Miocene likely facilitated the local adaptation of ancestral populations in Central China, the Hengduan Mountains, and the Yunnan–Guizhou Plateau, driving their expansion and diversification towards the west and south. Key environmental variables, including extreme temperature, temperature and precipitation variability, light, and altitude, were identified as major drivers of current niche divergence. Both niche conservatism and divergence were observed, with early conservatism followed by recent divergence. The Southern clade exhibits high heat and moisture tolerance, suggesting an adaptive shift, while the Central and Western clades retain ancestral drought and cold tolerance, displaying significant phylogenetic niche conservatism (PNC). We recommend prioritizing the conservation of *T. qinlingensis*, which exhibits the highest PNC level, particularly in the Qinling, Daba, and Taihang Mountains, which are highly degraded and vulnerable to future climate fluctuations.

## 1. Introduction

A key goal in biogeography and evolutionary biology is to utilize niche evolution dynamics to elucidate the mechanisms driving lineage diversification [1,2,3]. Niche divergence and phylogenetic niche conservatism (PNC) offer valuable insights into how lineages adapt to environmental changes, driving speciation. Niche divergence occurs when populations shift from their ancestral niches, occupying novel environmental systems along one or more axes in a multidimensional niche space [4,5]. This shift leads to ecological differentiation, which can contribute to lineage diversification [6,7]. Simultaneously, niche adaptation may also facilitate species divergence through PNC, where closely related species retain ancestral ecological requirements over time [8]. Under the PNC framework, ancient geological and climatic events fragmented ancestral ranges, isolating populations into distinct geographic or altitudinal units, potentially limiting gene flow and promoting diversification [5,9]. The interaction between niche divergence and conservatism shapes lineage divergence and influences niche similarity among members of the current clade [10]. Whether through divergence or conservatism, if closely related species occupy similar or identical niche spaces, it suggests that ecological factors may be secondary, with geographic isolation playing a more prominent role in driving divergence [11,12]. Consequently, the observed ecological niche similarities at the environmental and geographic spatial scales of closely related species provide critical insights into how geographic and environmental factors influence speciation and distribution [13]. Understanding ecological niche similarity and evolutionary dynamics is crucial for interpreting biogeographic history and guiding biodiversity conservation efforts.

Mountains could be considered biodiversity hotspots since they host nearly half of the world’s species [14]. Their diverse microclimates and local conditions foster unique genetic structures during ecological speciation [15]. Both rapid orogeny and glaciations have played key roles in isolating mountain species within distinct geographic units, limiting gene flow and promoting allopatric and parapatric speciation without requiring niche shifts [16,17]. While orogeny creates long-term geographic barriers, glaciations, which occur on shorter timescales, have likely been even more influential in driving species diversification for mountain species [18]. Additionally, these events can drive adaptive radiation, confining early species to ancestral ecological conditions, which may result in cryptic species. For example, the *Pinus armandii* and *Pedicularis siphonantha* complexes formed on the Qinghai–Tibet Plateau (QTP) exhibit phenotypic conservation despite distinct evolutionary pathways [19,20]. However, recent studies challenge adaptive radiation models for cryptic species, emphasizing the role of specialized niches in species formation beyond genetic constraints [21,22]. Cryptic species with distinct niche structures offer critical insights into species evolution, niche ecology, and climate change responses [23]. In this context, enhancing the understanding of ecological niche divergence and evolutionary dynamics in cryptic species and their closely related species is essential. Given the threats posed by global climate change and habitat loss, the biodiversity of cryptic species, currently overlooked, will face significant risks in the future [24,25]. However, research on the ecological niche characteristics and formation mechanisms of cryptic species within montane plant groups remains limited.

As a Tertiary relict gymnosperm, *Taxus* provides an important case for studying montane species’ niche evolution and lineage diversification formation mechanisms. Originating in the Early Cretaceous, *Taxus* underwent rapid diversification during the Late Miocene to Early Pliocene, resulting in extant species across Eurasia [26]. This genus possesses significant medicinal value, as its bark and leaves contain paclitaxel, one of the most widely used anti-cancer drugs in clinical treatment [27]. According to the latest assessment by the IUCN [28], nearly all *Taxus* species are classified as vulnerable (VU) or endangered (EN) due to historical overharvesting and land-use changes, with some, such as *T. floridana*, even listed as critically endangered (CR). However, its high phenotypic plasticity and complex evolutionary history have complicated its taxonomy, with classifications ranging from a single species [29] to 24 species and 55 varieties [30]. The taxonomic status and phylogenetic relationships of East Asian *Taxus* species, in particular, remain a subject of intense debate, significantly hindering conservation efforts [31,32,33]. Recent molecular barcoding studies have identified 16 global *Taxus* species, 11 of which mainly occur in East Asia: *T. wallichiana*, *T. florinii*, *T. contorta*, *T. cuspidata*, *T. chinensis*, *T. calcicola*, *T. phytonii*, *T. mairei*, and *T. qinlingensis* (a newly described species), as well as the Emei type and the Huangshan type [26,33,34,35]. The Emei and Huangshan types lack distinct phenotypic features (both are phenotypically similar to *T. chinensis*) but are phylogenetically monophyletic and were previously classified as cryptic species [36,37]. In East Asia mountains, 11 *Taxus* lineages exhibit a broad distribution, spanning from the Hindu Kush in the west to the Jiangnan Hills and Taiwan Mountains in the east, and from the Outer Khingan Range in the north to the Cordillera Central of the Philippines and the Greater Sunda Islands in the south (Figure 1a). These lineages or cryptic species inhabit tropical to subarctic ecological niches at elevations ranging from 100 to 3500 m, exhibiting a disjunct distribution pattern [33,38]. Despite this wide distribution, recent integrative taxonomic studies have primarily focused on the ecological niche similarities of a few closely related *Taxus* lineages [33,35,39]. Since the Tertiary period, tectonic activity, topographic evolution, and climate change have significantly influenced the habitat of 11 *Taxus* in East Asia, particularly the uplift of the Himalayas and the QTP due to the collision between the Indian and Eurasian plates [40]. These events facilitated montane plant migration, niche shifts, and species divergence across the region [41,42]. However, the ecological niche similarity among 11 East Asian *Taxus* lineages and cryptic species, the spatiotemporal patterns of their niche diversification, and the underlying ecogeographical mechanisms remain poorly understood. This gap hinders our understanding of their evolutionary history and the development of effective conservation strategies.

Phyloclimatic modeling, which integrates ecological niche models with phylogenetic data, provides insights into niche evolution and lineage trajectories [43]. This approach combines multiple analytical frameworks, such as reconstructing ancestral niche breadth, ecological tolerances, and niche trait disparity over time. It has been widely applied to study niche evolution in diverse species, including *Scutiger boulengeri* [1], *Viperidae* [44], and *Abies* [45]. Using occurrence, environmental, and reconstructed phylogenetic data, we applied multiple ecological niche and phyloclimatic modeling approaches to examine the ecological niche characteristics and evolutionary patterns of East Asian *Taxus* lineages, while identifying conservation priorities. Our objectives are (1) to reveal the ecological niche similarity and evolution patterns of 11 *Taxus* lineages in geographic and environmental space; (2) to explore the role of geographic (including paleogeoclimatic events) and ecological factors in niche divergence and evolution among lineages; and (3) to evaluate PNC levels to inform conservation strategies.

## 2. Results

### 2.1. Phylogenetic Relationships and Divergence Time

The lengths of individual sequences for the 13 cpDNA, ITS, and NEEDLY regions, as well as the concatenated sequences, were 11,914, 1978, 1500, and 15,392 bp, respectively. Bayesian inference revealed that the posterior probabilities for 10 nodes ranged from 0.75 to 1, indicating higher support (Figure A1; Table A1). The dated phylogenetic tree for East Asian *Taxus* lineages revealed two major groups that diverged around 8.49 million ago (Ma) [95% highest posterior density (HPD) = 7.44–10.67; Figure 1c and Figure A1 and Table A1]. The first group, representing the Western and Northern clades, includes *T. wallichiana*, *T. florinii*, *T. contorta*, and *T. cuspidata*, and split around 7.07 Ma (HPD = 6.18–7.92). The lineages within these clades diverged between 5.95 Ma (HPD = 5.09–6.78) and 5.02 Ma (HPD = 5.77–4.18). The second group exhibits significant divergence, with a split between *T. qinlingensis* and the other lineages in the Central and Southern clades around 6.73 Ma (HPD = 5.79–8.63). The Southern clade diverged from *T. chinensis* and the Emei type at 6.05 Ma (HPD = 5.23–6.83), with further divergence within the Southern clade estimated at 4.73 Ma (HPD = 3.56–5.44), 3.42 Ma (HPD = 2.77–3.91), and 2.3 Ma (HPD = 1.88–2.57; Figure 1c and Figure A1 and Table A1).

### 2.2. Current Geographical Spatial Niche Comparison

The average TSS and AUC values for models including ANN, CTA, FDA, GBM, GLM, MAXENT, and RF were above 0.75, while GAM and SRE did not meet the integration criteria. *T. wallichiana*, *T. chinensis*, and *T. qinlingensis* displayed higher TSS and AUC values, indicating stronger simulations and consistent evaluations (Figure A2). The Western clade primarily inhabits the Hengduan Mountains and the southern and eastern QTP (Figure 2a–c), while the Northern clade mainly occupies the southwestern Siberian Plateau, the Sikhote-Alin Mountains, the Gaymard Plateau, and the southern Sakhalin Island (Figure 2d). The Central clade is concentrated in the Qinling, Daba, and Taihang Mountains, and around the Sichuan Basin (Figure 2e–g). The Southern clade shows the greatest distributional differences, with the Huangshan type thriving in the eastern Jiangnan Hills and southern Japan, *T. calcicola* in the Yunnan and Guizhou Plateau, *T. phytonii* across various islands and coastal areas of East and South Asia, and *T. mairei* ranging from the Daba to Nanling Mountains, the Hengduan Mountains, southern QTP, and southern Japan (Figure 2h–k). The α-diversity centers of yew lineages in East Asia are mainly concentrated in the Hengduan Mountains, the western Sichuan Basin, and the Yunnan–Guizhou Plateau (Figure 2l). Among the lineages, *T. mairei* occupies the largest potential suitable area, while the Huangshan lineage occupies the smallest (Table 1). From the Current to 2070 and 2070 to 2090, the southern and southwestern Tibetan Plateau, the northwestern Sichuan Basin, southeastern China, and Taiwan Island all show greater climate variability (Figure 6).

The annual precipitation across the 11 yew lineages ranges from 41.4 to 470.6 cm, the mean annual temperature ranges from −1.20 to 27 °C, and the altitude ranges from 294 to 3335 m (Appendix A). The Whittaker biome plot indicates that the yew lineages primarily occupy seven community ecological niches, with temperate seasonal forest (65.54%) and woodland/shrubland (21.15%) being the most common (Figure A3). At the clade level, the Western clade (27.36%) and Central clade (38.3%) are slightly biased towards woodland/shrubland, the Northern clade is biased towards boreal forest (38.1%), and the Southern clade shows the greatest community niche diversity, occupying temperate and tropical forests (Figure A3; Appendix A).

### 2.3. Current Environmental Space Niche Comparison

The first two principal components of the environmental principle components analysis (PCA-env) explained 70.84% of the environmental variance (PC1: 39.62%; PC2: 31.22%) (Figure 3a). Low spatial overlap was observed between the Southern clade, Northern clade, and the other clades, while the Central clade and Western clade exhibited high overlap (Figure 3a). Schoener’s *D* analysis showed that 52.72% of paired comparisons had limited niche overlap (0–0.2), 34.55% had low overlap, 9.09% had moderate overlap, and 3.64% had high overlap, primarily between the Central and Western clades, such as *T. chinensis* VS. the Emei type and *T. mairei*, and *T. contorta* VS. *T. qinlingensis* and the Emei type, and *T. wallichiana* VS. *T. qinlingensis* and *T. contorta* (Figure 3b). Similar to the trend observed in the geographic space’s potential suitable area, *T. mairei* exhibited the broadest niche breadth, while the Huangshan lineage had the narrowest (Table 1).

In the PCA-env niche identity and similarity tests, 80% of the paired comparisons supported niche conservatism. Among them, 13 pairs (23.64%) and 15 pairs (27.27%) showed significant niche conservatism (*p* < 0.05) (Figure A4). The ENMtools v1.0-based niche identity and background similarity tests show that 54.55% and 58.18% of the paired comparisons supported niche conservatism, respectively (Figure A5). Overall, the null hypothesis tests for both approaches revealed that niche divergence predominantly occurred between the Southern clade and the other lineages, while niche conservatism was primarily observed between the Central and Western clades (Figure A4 and Figure A5).

### 2.4. Environmental Drivers of Current Niche Patterns

The ensemble ecological niche model (eENM) identified key environmental variables influencing niche geographical space, including the mean temperature of the coldest quarter (bio11), the temperature annual range (bio7), elevation (EL), and solar radiation (SRAD; Figure A6). The Western clade is most influenced by EL and bio11, the Northern by the max temperature of the warmest month (bio5) and annual precipitation (bio12), the Central by bio11 and bio7, and the Southern by bio7 and the mean diurnal range (bio2) (Figure A6). The PCA-env analysis revealed the environmental variables shaping the niche space: PC1 was predominantly represented by the precipitation of the driest quarter (bio17), precipitation seasonality (bio15), and bio12, while PC2 was mainly explained by bio5, bio7, and bio2. Notably, bio2 and bio15 were key factors explaining environmental preferences between the Southern lineage and the other lineages (Table A2). The density plots demonstrated distinct habitat differences among the 11 yew lineages, particularly in bio17, SRAD, and EL (Figure A7).

### 2.5. Ancestral Area Reconstruction (AAR)

The AAR identified the Bayesian Binary Markov chain Monte Carlo (BBM) model as the best fit (Table A3). The ancestral reconstruction based on the BBM model suggests that East Asian yew speciation primarily occurred in Central China, the Hengduan Mountains, the Yunnan–Guizhou–Guangxi region, and the Indo-China Peninsula. Three key historic events were identified, including (i) dispersal from the Hengduan Mountains to the Qinghai–Tibetan plateau and Hindu Kush mountains (probability = 0.77); (ii) dispersal from Central China to Eastern and Southern China, and the Yunnan–Guizhou–Guangxi region (probability = 0.44); and (iii) dispersal from the Yunnan–Guizhou–Guangxi region to Taiwan and the Philippines (probability = 0.58; Table A1; Figure 1c).

### 2.6. Ancestral State Estimation (ASE)

Eight environmental variables showed significant phylogenetic signals: Bio2, Bio5, Bio11, Bio12, Bio17, EL, SRAD, and TBS (topsoil base saturation). Both Pagel’s λ (*p* < 0.01) and Blomberg’s K (*p* < 0.05) supported these findings. The Brownian motion (BM) model best explained the evolution of these variables (ΔAIC > 10) (Table 2). The ancestral state estimates based on the BM model suggest both phylogenetic niche conservatism and evolutionary divergence in the East Asian yew lineage (Figure 4). The clades show distinct environmental preferences, with the Western clade favoring higher elevations and lower precipitation seasonality, the Northern clade favoring lower elevations with higher precipitation seasonality, the Central clade favoring lower temperatures, and the Southern clade favoring warmer, wetter conditions. The Central clade shows the highest PNC levels, with *T. chinensis* and *T. qinlingensis* having seven environmental variables similar to their ancestors. In contrast, the Southern clade exhibits the lowest PNC levels, particularly *T. mairei* and *T. calcicola*, with only one shared environmental variable (Figure 4).

### 2.7. Predicted Niche Occupancy (PNO) and Ancestral Tolerances

The PNO profiles generated by the eENM reveal heterogeneity among the yew lineages, especially between the Southern clade and the others (Figure 5 and Figure A8). The Southern clade shows higher peak suitability across environmental variables, with the Huangshan type having the highest extreme values for Bio2, Bio7, Bio15, Bio17, and SRAD (Figure A8). The ancestral tolerance reconstruction highlights key differences between the Southern clade and the others, particularly in Bio5, Bio12, Bio17, and TBS (Figure 5). The Southern clade prefers environments with higher temperatures and precipitation, while the other clades favor cooler, drier conditions (Figure 5 and Figure A8). Additionally, the lineages from different evolutionary clades overlap significantly in several environmental variables, especially Bio7, Bio11, Bio15, and SRAD, which show narrower tolerance densities. Notably, strong niche conservatism is observed between *T. contorta* and *T. cuspidata* across almost all 10 environmental factors, despite their wide geographic separation. Similar patterns are found in other clades (e.g., *T. wallichiana* and *T. florinii*, *T. chinensis* and *T. qinlingensis* and the Emei type), but they were all geographically closer (Figure 5).

The disparity through time (DTT) analyses show that almost all environmental factors had zero divergence at deep nodes, indicating early conservatism in major evolutionary clades (Figure A9). While most environmental factors did not exceed the 95% confidence interval (CI) of the zero BM model, some peaks suggested slight divergence at recent nodes and mild evolution within clades, reflected in the positive morphological disparity index (MDI). In the recent past, EL levels exceeded the 95% CI for zero species formation. Overall, higher DTT differences across environmental factors were concentrated in more recent evolutionary clades (relative times 0.3–0.6). In bio2, bio5, bio7, bio11, bio17, SRAD, and EL, the MDI values of the trees were all positive; however, the values ranged from 0.00124 to 0.213, suggesting that ecological niche evolution within clades was at a low level and that the ecological niches between clades were somewhat conserved, with most of the low ecological differences originating from more recent divergence events. The other three environmental factors (Bio12, Bio15, and TBS) had lower total within-clade differences than expected from the BM model for most of the phylogenetic tree, and their negative MDI values (−0.016, −0.003, and −0.007, respectively) suggest that their differences occurred mainly among the major lineages, whereas the lineages were ecologically niche-conservative within the lineages, and that they arose in the early phylogenetic stages (Figure A9).

## 3. Discussion

### 3.1. Ecological Niche Characterization

Understanding the ecological niche similarity characteristics among closely related species is crucial for clarifying the relative contribution of geographic and ecological factors to species divergence [46]. In this study, in both geographical space and environmental space, the Central and Western clades exhibit the most similar ecological niches, while the Northern and Southern clades are more distinct (Figure 2 and Figure 3). The differences in the actual ecological niches and background environments between the two taxa determine the extent of niche divergence between them [47]. Generally, East Asian yew lineages are found in the understory or at the forest margins of evergreen broad-leaved forests, coniferous forests, or mixed forests. Specifically, the Northern clade occurs in lowland to montane regions with acidic soils; the Central and Western clades are mainly found in humid valleys, riversides, and forest edges; and the Southern clade is concentrated on slopes and mountain tops [33,34,38]. In terms of the mean annual temperature and precipitation, the Central (9.86–11.95 °C; 82.06–113.36 cm) and Western (8.65–11.78 °C; 82.73–130.41 cm) clades are more similar to each other and differ significantly from the Southern (15.58–18.08 °C; 140.22–281.64 cm) and Northern (3.66 °C; 86.49 cm) clades (Appendix A). In addition, the Southern clade shows the greatest degree of niche divergence from the other clades (Figure 2 and Figure 3). On the one hand, the Southern clade, except for *T. mairei*, is distributed in an island-like pattern, either on karst landscapes or actual islands [33,48]. On the other hand, the Southern clade’s biome is the most diverse, spanning from the subtropical to tropical zones, which is reflected in the precipitation variability and topsoil base saturation among the four lineages within the clade (Figure A3; Appendix A). The isolated habitats, combined with unique soil and precipitation conditions, are likely driven by the geographic and environmental niche divergence within the Southern clade. The null hypothesis tests also reveal significant niche divergence among lineages within the Southern clade and the others (Figure A4 and Figure A5).

Niche overlap is prominent between certain closely related species pairs in the Central and Western clades, including *T. chinensis* vs. the Emei type, *T. mairei*, *T. contorta* vs. *T. qinlingensis* and the Emei type, and *T. wallichiana* vs. *T. qinlingensis* and *T. contorta* (Figure 2 and Figure 3). This suggests that ecological divergence alone may not be the major driving force behind these lineages’ formation. Ecological speciation theory proposes that reproductive isolation between populations evolves through eco-based divergent selection [49]. However, our results suggest higher PNC in the Central and Western lineages, indicating that these lineages have retained similar ecological niches throughout their evolutionary history (Figure 4, Figure 5, Figure A4 and Figure A5). The existence of PNC could promote speciation when species fail to adapt to new ecological conditions [18]. In this mechanism, geographic ranges become fragmented, and allopatric divergence occurs through isolation [5]. Additionally, these lineages are predominantly distributed around the Sichuan Basin, benefiting from the region’s continuous mountains and forests, which offer refuges for independent evolution, migration corridors, and opportunities for hybridization [26,37]. Previous studies have shown that these lineages exhibit higher phenotypic plasticity and more complex phylogenetic relationships compared to others [38,50]. Consequently, we speculate that isolation and gene flow events, along with similar population histories, may explain the observed high niche overlap between these lineages. Although the eENM and Whittaker biome classification suggests core distributions for these two cryptic species in geographical space (Figure 2, Appendix A), Schoener’s *D* analysis reveals that the Emei type exhibits significant environmental niche overlap with *T. contorta* and *T. chinensis*, while the Huangshan type maintains a more distinct environmental niche from the other lineages (Figure 3). Super barcodes indicate that the Emei type is closely related to *T. wallichiana* and the Huangshan type to *T. chinensis* and *T. florinii* [34]. Multiple molecular markers suggest that the Emei type emerged from hybridization between *T. chinensis* and *T. wallichiana* and the Huangshan type arose from the migration and adaptation of *T. chinensis* [26]. Chloroplast genome data support the monophyly of the Emei type, with the Huangshan type being more closely related to *T. contorta* [51]. These findings highlight the need for more advanced sequencing and analytical approaches to fully understand the phylogenetic relationship of these two cryptic species with other lineages, given the complexities of gene marker coverage, hybridization, and incomplete lineage sorting. Nonetheless, we contend that long-term gene flow, isolation, and adaptation have likely played an important role in the formation of cryptic species in *Taxus*. For instance, the formation of the Emei type with *T. florinii* is hypothesized to result from bidirectional chloroplast capture, long-term gene flow, and isolation [39]. Meanwhile, ecological niche evolutionary divergence and phylogenetic niche conservatism likely played crucial roles in the maintenance of cryptic species in the Central and Southern clades, respectively. The Emei type demonstrates stronger phylogenetic niche conservatism (Figure 4). Given the generally slow rate of ecological niche evolution in *Taxus* in East Asia (Figure A9), this suggests that long-term gene flow barriers caused by geographic isolation may play a significant role in the speciation process of the Emei type. In contrast, the Huangshan type’s niche overlap with other lineages is limited (Figure 2 and Figure 3), and it shows greater tolerance to temperature, precipitation, and light variables (Figure 5 and Figure A8), suggesting that its unique niche structure is primarily driven by ecological differentiation.

### 3.2. Evolutionary History and Biogeographic Distribution

Inferring species range shifts is crucial for understanding how evolutionary lineages deviate from their ancestral niches to occupy different ecological spaces. Based on the reconstructed phylogenetic tree and geographical distribution characteristics, the 11 *Taxus* lineages in East Asia are grouped into four clades (Figure 1 and Figure A1). Significant divergence among the four clades was estimated to have been initiated in the Late Miocene to Early Pliocene (Figure 1; Table A1). Central China, the Hengduan Mountains, and the Yunnan–Guizhou Plateau were identified as the primary speciation areas for East Asian *Taxus* lineages (Figure 1; Table A1). Orogenic movements in the Tibet–Himalaya–Hengduan Mountains region, combined with the Indian Ocean and East Asian monsoons that brought increased precipitation, isolated ancestral alpine plant groups during the early to Mid-Miocene [52]. During the Late Miocene to Early Pliocene, the continued and rapid uplift of the Himalayas–Hengduan Mountains, along with sharp temperature drops and increased aridity, enhanced the isolation and divergence of these lineages [53]. The Yunnan–Guizhou Plateau, Sichuan Basin, and neighboring regions have been identified as long-term refugia for East Asian relict plants [54]. These regions, characterized by numerous mountains and valleys that remained unglaciated during the Quaternary glacial periods, provided stable habitats and genetic exchange hubs, further stimulating the diversification of various East Asian Tertiary relict genera such *Craigia*, *Dipelta*, and *Davidia* [41]. Comparing fossil records, we hypothesize that since the Miocene, East Asian *Taxus* has likely undergone southward contraction into refugia due to orogenic movements and cold, arid climates (Figure 1). This may have led to isolation and introgressive hybridization in the identified species differentiation areas. For instance, the cooling and drying effects induced by the uplift of QTP during the Late Miocene may have led to the isolation and diversification of the *T. wallichiana* complex in fragmented forests [55,56]. DTT analysis also shows that early phylogenetic conservatism among East Asian *Taxus* lineages was mainly driven by changes in precipitation seasonality and topsoil saturation (Figure A9). Furthermore, the cyclical appearance and disappearance of geographic and ecological barriers in the identified speciation regions due to Pleistocene glacial–interglacial cycles likely repeatedly interrupted and connected gene flow among ancestral *Taxus* lineages, further promoting the genus’s contraction, expansion, and diversification [57,58]. The eENM results also highlight the highest current *Taxus* lineage diversity in Central China, the Hengduan Mountains, and the Yunnan–Guizhou Plateau (Figure 2).

Multiple migrations have played a key role in distinguishing the biogeographic history of Northern Hemisphere conifers from that of Southern Hemisphere conifers [59,60]. This study identifies multiple migration events for East Asian *Taxus* lineages since the Late Miocene (Figure 1; Table A1). Unlike other narrow-range “living fossil” species, the *Taxus* genus remains widely distributed across diverse environmental conditions, from oceanic to continental and Mediterranean climates in North America and Eurasia [39,61], reflecting its remarkable migration capabilities. Three key migration routes were identified in this study (Figure 1; Table A1). The first involved ancestral lineages from Central China and the Hengduan Mountains migrating to the Tibetan Plateau and Hindu Kush during the Miocene–Pliocene (3.98–6.06 Ma), consistent with Möller’s hypothesis on the migration and diversification of *T. wallichiana* [26]. Simultaneously, central ancestral lineages spread to Eastern, Southern, and southwestern China, possibly leading to the speciation of *T. chinensis*, *T. qinlingensis*, and the Huangshan type. The higher ecological niche overlap between *T. chinensis* and *T. qinlingensis* (Figure 2 and Figure 3), coupled with their close phylogenetic relationships with the Huangshan type [34,51], suggest likely isolation and localized adaptation during migration. The final route involved the migration of southern and eastern Chinese lineages to the Yunnan–Guizhou Plateau, Taiwan, and the Philippine Islands during the late Pliocene to early Pleistocene (1.51–3.01 Ma). Most of these lineages are located along bird migration routes in the East Asia–Australia and West Pacific flyways, with frugivorous birds, such as *Turdus* species, likely facilitating long-distance dispersal [62]. Mechanisms such as seed introgression and sex ratio adjustment (SRAMs) may have further driven genetic divergence and local adaptation in these Southern lineages [63]. In addition, previous studies indicate that during the late Miocene (~7 Ma), the Eurasian *Taxus* ancestor diverged into northern Eurasian lineages, which subsequently migrated to Europe to form *T. baccata*, to the Himalayas to form *T. contorta*, and to Northeast Asia to form *T. cuspidata* [26]. Although the biogeographic history of the northern group (*T. cuspidata*) is not fully reconstructed, the higher PNC levels between *T. contorta* and *T. cuspidata*, along with continuous fossil records from Northeast Asia, North China, and Central Asia, support the above inference (Figure 1, Figure 4, and Figure 5). In addition, the fossil records from interglacial periods in the European steppe region, as depicted in the circumpolar distribution map of *Taxus*, also underscore the close phylogenetic relationship among *T. baccata*, *T. contorta*, and *T. cuspidata* [64].

### 3.3. Environmental Drivers of Ecological Niche Divergence and Evolution

Plant ecological niche diversification is typically influenced by multiple environmental factors [65]. For the *Taxus* lineages across East Asia, key environmental variables, such as extreme temperatures, temperature and precipitation variability, light, and altitude, have been identified as significant factors influencing current niche divergence in geographical and ecological space (Figure A6 and Figure A7; Table A2). Moderate high temperatures enhance physiological activity and photosynthesis in yew during the growing season, which supports dormancy and growth, while extreme cold impairs these processes [66]. Although the yew is drought-tolerant, it is sensitive to waterlogging [67]. Dormant and growing seasons, as well as diurnal precipitation variations, affect chlorophyll content in the yew, leading to physiological changes that are reflected in radial growth [68]. As a shade-tolerant species typically growing in the 2nd–3rd layers of forest canopies, excessive canopy closure limits the yew’s growth potential [69]. Additionally, environmental differences across elevations can influence phenology, affecting population dynamics and distribution [66,68]. The combined effects of temperature, precipitation, light, and altitudinal gradients not only constrain the physiological behavior of East Asian *Taxus* lineages but also influence their growth characteristics, dispersal directions, and reproductive cycles, ultimately leading to ecological niche diversification and spatial differentiation among lineages.

Phylogenetic signals of environmental variables are essential for understanding plants’ long-term adaptation and evolutionary patterns. When analyzing the PS for individual variables, most were best modeled by a BM evolutionary model (Table 2), indicating a phylogenetic random walk. However, caution is advised as high PS does not always equate to conservatism [70]. Among the seven environmental variables related to climate, light, and soil, we observed K values greater than 1, suggesting greater similarity to ancestors under BM, possibly due to evolutionary constraints or stabilizing selection [70]. The slow evolutionary rates common among conifers, coupled with their highly conserved genome collinearity and long generation times, may lead to decoupled gene evolution from environmental changes, resulting in “conservative” niche traits [71]. The ASE analyses also revealed a complex evolutionary pattern characterized by both ecological niche conservatism and divergence in East Asian yew lineages (Figure 4). Notably, the PCA-env, environmental tolerance, and PNO analyses indicated that climatic factors primarily drove the ecological niche divergences between the Northern, Central, Western, and Southern clades, with the former favoring drier, colder environments, while the latter preferred warmer, wetter conditions (Figure 3, Figure 5 and Figure A8). The biomes of the Northern, Central, Western, and Southern clades transition from cold-temperate subalpine coniferous forests to temperate shrublands, tropical rainforests, and subtropical rainforests, respectively (Figure 2 and Figure A3). The DTT analysis revealed early niche conservatism and recent divergence linked to temperature variation, extreme temperatures, precipitation seasonality, light, and elevation (Figure A9). Thus, we hypothesize that the drought and cold tolerance observed in the Northern, Central, and Western clades are likely the result of early adaptation to geologic and climatic changes, coupled with ongoing local adaptation within these clade lineages. This is evidenced by variations in leaf and spongy tissue thickness for these lineages [38]. Given the relatively recent divergence of the Southern clade, its high tolerance to heat and humidity is likely a key adaptive innovation for coping with global climate change. Similar genus-level niche adaptation gradients have been observed in dendritic frogs [72], North American salamanders [73], and *Cedrela* [74]. Nevertheless, more detailed phylogeographic and phenotype–physiology–genetics–environment association studies using broader genomic data are needed to fully explore the niche adaptive evolution of East Asian *Taxus* lineages.

### 3.4. PNC Pattern and Conservation Revelations

The repeated fragmentation–reconnection cycles since the Late Miocene climate cycles likely drove the divergence and speciation within East Asian *Taxus*. Geological and environmental changes, regional ecological heterogeneity, and genetic mixing characteristics provided opportunities for *Taxus* lineages to diversify across a wide range of geographic and ecological settings in East Asia. The spatiotemporal variations highlighted both the similarities and differences in niche evolution patterns among these lineages. For the Southern *Taxus* lineages, recent speciation events and their widespread adaptation to high temperature and humidity may have facilitated their ecological shifts and diversification into subtropical and tropical niches, ultimately deepening niche divergence among them and other lineages. In contrast, for the Central and Western *Taxus* lineages, early rapid geological and climatic changes caused isolation, and the ecological constraints of the ancestral niche (dry and cold environments) likely accelerated their phylogenetic conservatism, thus emphasizing post-speciation niche conservatism patterns. These phylogenetically conservative lineages overlap in the glacial refugia of the Hengduan and Qinling–Daba mountain ranges during the Quaternary [75]. The ASE and environmental tolerance, PNO, and DTT analyses all also indicate that Central and Western clades exhibit slower niche evolution rates and are likely adapted to similar environmental gradients (Figure 4, Figure 5, Figure A8 and Figure A9).

The concept of cryptic species integrates evolutionary biology, taxonomy, and conservation planning, and recognizing these distinct taxa and their geographic distribution patterns is crucial for understanding their biogeography and speciation, as well as for tailoring conservation strategies [76]. Using East Asian *Taxus* lineages as a prominent model for Tertiary relict “living fossil” species, we integrated various ecological niche modeling methods and conservation goals to explore the group’s niche characterization, biogeographic history, and conservation priorities. Given their unique ecological niche evolutionary traits, biogeographic history, and limited regeneration capacity, we argue that all isolated *Taxus* populations should receive priority conservation. Special attention should be given to cryptic species, such as the Emei type and Huangshan type, which exhibit narrow niche widths and restricted distribution areas (Table 1). The Emei type is distinguished by its higher niche conservatism, whereas the Huangshan type is characterized by a narrow niche tolerance range and a unique ecological niche structure (Figure 4, Figure 5 and Figure A8). Future research should prioritize investigating the genetic variation patterns and population dynamics of these cryptic species to better understand their ecological and evolutionary significance.

Long-lived conifers with slow niche evolution rates are typically more susceptible to population declines and climatic fluctuations than faster-adapting shrubs and herbaceous species [77]. However, the central and western core distribution areas, including the southern and southwestern Tibetan Plateau, northwestern Sichuan Basin, southeastern China, and Taiwan Island are predicted to experience greater climatic fluctuations in the future (Figure 6c,d). Furthermore, significant conservation gaps remain in the southern Himalayas, Hengduan, and the Qinling–Daba Mountains region for these lineages (Figure 6b). Therefore, we recommend focusing conservation efforts on regions with significant climate fluctuations and protection gaps in the Central and Western lineages with higher PNC. Most urgently, based on our field surveys, the distribution area of wild *T. qinlingensi* has gradually decreased due to climate change, ecological degradation, and human activities (e.g., illegal logging, infrastructure construction), showing a clear trend of fragmentation and even sparse distribution along roadsides, village peripheries, and landfill sites (Figure 6a). Furthermore, the total natural population of *T. qinlingensis* is fewer than 30,000 individuals (Wu et al., 2024), with low genotype diversity (unpublished data) and high niche conservatism (Figure 4), reflecting its limited adaptability to environmental changes. Large-scale habitat reduction and fragmentation have intensified lineage isolation, restricted gene flow, and increased the chances of genetic drift and inbreeding, thereby putting the small populations of *T. qinlingensis* at a higher risk of extinction (unpublished data). The decline in population size, fragmented distribution, low genetic variation, and limited adaptive capacity pose significant challenges to the conservation of *T. qinlingensis*. Recently, the taxonomic status of the *T. qinlingensis* “species” has been established [35]. Therefore, we recommend a comprehensive reassessment of the endangered status of *T. qinlingensis*, including naturally distributed populations in the western Sichuan Basin and the Yunnan–Guizhou Plateau, with the aim of elevating its conservation status as a distinct species. Given the ongoing threats, we further suggest enhancing public awareness campaigns and strengthening monitoring systems, including the use of satellite surveillance and on-ground patrolling, to mitigate illegal logging activities. Additionally, expanding fragmented reserves, particularly in the Qinling–Daba and Taihang Mountains region, and implementing adaptive conservation strategies, like assisted migration and mitigating overdevelopment and habitat destruction. Future systematic surveys, extensive sampling, and the application of whole-genome sequencing technologies will enable more detailed conservation planning for East Asian *Taxus* based on spatial genetic structure, adaptive evolutionary patterns, and genomic vulnerabilities.

## 4. Materials and Methods

### 4.1. Data Collection

To reconstruct the phylogenetic tree of yews across East Asia, we selected one representative sample per lineage (11 *Taxus* lineages) plus two outgroups (13 samples in total) from the NCBI GenBank database. We analyzed 13 chloroplast DNA (cpDNA) markers, including *mat*K, *ccs*A, *rpo*A, *rpo*B, *rpo*C1, *cem*A, *rbc*L, *trnf*m-S, *psb*K-*psb*I, *trn*S-R, *rpl*16, *trn*L-F, and *trn*H-*psb*A, and two nuclear markers (ITS and NEEDLY) (Table A4). Sequence alignment and editing were conducted using BioEdit v7.0 [78] and MEGA X [79].

Species distribution data were gathered from our fieldwork and the relevant literature after 2013. To enhance accuracy, we prioritized distribution points verified through DNA barcoding. Initially, 528 distribution points were collected. Following the large-scale modeling screening criteria of Wu et al. [80], we used the “spThin v0.2.0” R package [81] to remove duplicates and ambiguous records, retaining one record per 2.5′ × 2.5′ grid cell. This process resulted in a final dataset of 383 unique spatial records representing 11 yew lineages across East Asia: *T. wallichiana* (n = 54), *T. florinii* (n = 20), *T. contorta* (n = 32), *T. cuspidata* (n = 42), *T. chinensis* (n = 34), *T. calcicola* (n = 13), *T. phytonii* (n = 14), *T. mairei* (n = 104), *T. qinlingensis* (n = 45), the Huangshan type (n = 5), and the Emei type (n = 18) (Figure 1a, Appendix A). Additionally, we collected nine fossil record points to help elucidate the biogeographic history of the 11 *Taxus* lineages (Figure 1a, Appendix A).

We obtained 41 environmental variables, including climate, solar radiation, topography, biological interactions, and soil data, all at a uniform resolution of 2.5 arc minutes. To prevent overfitting of the environmental factors, we performed Pearson correlation analysis on the 41 variables using the R package “Hmisc v5.1-3” [82]. Variables with a high correlation (|r| > 0.75) were excluded, resulting in 16 variables for the subsequent ecological niche analysis of the 11 yew lineages (Figure A10, Table 3). For climate fluctuation analysis, we acquired future climate data for the 2070s and 2090s from WorldClim v2.1, consistent with the current climate variables. Considering that the CMCC Earth System Model 2 (CMCC-ESM2) effectively captures global climate variability and change, and that the SSP2-4.5 scenario is more realistic than both overly pessimistic and overly optimistic scenarios, we selected this model and scenario for subsequent analysis [83].

### 4.2. Phylogenetic Reconstruction and Divergence Time Estimation

Tree construction followed the methodology of combining nuclear and plastid sequences, as outlined by Möller et al. [26], to minimize phylogenetic incongruence between nuclear and plastid markers. Incongruence length difference (ILD) and saturation tests were performed to assess the suitability of concatenating nuclear and plastid markers for phylogenetic reconstruction. First, the “phylotools v0.2.2” R package [84] was used to merge the 13 cpDNA markers, ITS, and NEEDLY fragments into a supermatrix. Next, ILD testing was conducted on the concatenated sequences using PAUP v4 software [85]; sequences were considered suitable for merging if *p* ≥ 0.05. Sequence substitution saturation was then assessed using DAMBE v5.2 software [86] to determine whether the concatenated sequences were appropriate for phylogenetic reconstruction. Datasets with substitution saturation (*Iss*) < critical one (*Iss.c*) and *p* < 0.05 were considered suitable for phylogenetic reconstruction. Since the results of the ILD and saturation tests for the concatenated sequences (13 cpDNA markers, ITS, and NEEDLY) indicated high compatibility (*p* = 0.18 > 0.05) and no substitution saturation (*Iss* < *Iss.c*, *p* = 0.000), the merged data were deemed suitable for subsequent analysis (Table A5).

The phylogenetic relationships of 11 *Taxus* lineages were reconstructed using Bayesian Markov chain Monte Carlo (MCMC) inference [87] with MrBayes v3.2.6 [88]. The optimal substitution model (GTR+G) for the combined sequences was determined using jModelTest v2 [89]. The MCMC analysis ran for 10,000,000 generations, sampling one tree every 1000 generations, with the first 10% discarded as burn-in. Convergence was assessed using Tracer v17 [90], with an effective sample size (ESS) threshold set to greater than 200 for each parameter.

Divergence times were estimated using BEAST v2.6.2 [91] under the lognormal relaxed clock model, with a Yule tree prior and the GTR+G substitution model. We utilized secondary calibration points based on fossil records from Möller et al. [26], including calibration points A and B with node ages of 93.14 and 151.58 Ma, respectively. Their 95% HPD intervals were [63.8, 122.1] and [77, 232.1] million years, both following a normal prior distribution (Figure A1; Table A6). The MCMC analysis ran for 20,000,000 generations, sampling one tree every 1000 generations. Tracer v1.7 [90] was used to check the convergence of ESS values, with a threshold set to greater than 200. The maximum clade credibility (MCC) tree and divergence times for each node, along with their corresponding 95% HPD intervals, were summarized using TreeAnnotator v1.8.2 [91]. The initial 10% of the data was discarded as burn-in. The resulting time tree was visualized and edited using FigTree v1.4.2 [92].

### 4.3. Current Geographical Spatial Niche Comparison

We applied an eENM and community ecological niche stratification to compare ecological niches in geographical space. To minimize bias from using a single model and the limited number of distribution points, we employed an integrated algorithm consisting of nine models, i.e., ANN, CTA, FDA, GAM, GBM, GLM, MAXENT, RF, and SRE, on the Biomod2 platform [93] to simulate the distributions of the 11 *Taxus* lineages in East Asia (Table A7). To improve the accuracy of the eENM, 10,000 pseudo-absence points were generated randomly across the yews’ distribution ranges [94]. Each model was trained using 75% of the presence data, with 25% reserved for testing. The models were evaluated based on the area under the curve (AUC) and true skill statistics (TSS). Only models with AUC > 0.8 and TSS > 0.75 were included in the ensemble analysis. Habitat suitability was categorized into four levels: unsuitable (0–0.2), low suitability (0.2–0.4), medium suitability (0.4–0.6), and high suitability (0.6–1.0) [95]. Binary presence–absence maps were generated using a threshold of 0.5 and combined in ArcGIS 10.5 [96] to create an *α*-diversity distribution map for East Asian yews. Conservation gaps were identified by overlaying these maps with 2021 Chinese nature reserve data. Principal component analysis (PCA) of the environment variable extract value was performed using the “ggbiplot” package [97] to obtain PC1 scores for each period, quantifying future climate fluctuation across yew habitats based on standardized differences between periods (|PC1_2070_ − PC1_Current_| and |PC1_2090_ − PC1_2070_|). To further elucidate the differences in community niches among *Taxus* lineages, we employed the “plotbiomes 0.0.0.9001” and “elevatr v0.99.0” R packages [98,99] to classify the geographical distribution points of the 11 *Taxus* lineages into Whittaker biomes. Community niches encompass both the ecosystem types in which species persist across various habitats and the ecological functional characteristics that these ecosystems provide to support species survival, resource acquisition, and population dynamics [100].

### 4.4. Current Environmental Space Niche Comparison

Principal component analysis of PCA-env was performed using the R package “ecospat v4.1.1” [101] to compare the niche similarity of the 11 *Taxus* lineages in environmental space and to explore the effects of environmental factors on niche divergence or conservatism [46]. Referring to Moreno-Contreras et al. [102], Schoener’s *D* index [103] was employed consistently to assess niche overlap, which was classified into five categories: none/very low overlap (0–0.2), low overlap (0.2–0.4), moderate overlap (0.4–0.6), high overlap (0.6–0.8), and very high overlap (0.8–1.0) [104].

Two null hypothesis methods were used to test niche divergence. The first method involved comparing the observed Schoener’s *D* values from PCA-env with the expected values generated by 1000 random iterations to test the identity and background similarity. The second method used species distribution and environmental data to perform niche identity and background similarity tests with the R package “ENMtools v1.0” [105], repeated 100 times. Both methods followed the same criteria: if the observed Schoener’s *D* values significantly deviated from the expected values (*p* < 0.05), niche divergence was indicated; otherwise, niche conservatism was inferred.

### 4.5. Ancestral Area Reconstruction

Ancestral distributions were reconstructed using RASP v4.0, applying Statistical Dispersal–Vicariance Analysis (S-DIVA), Statistical Dispersal–Extinction Cladogenesis (S-DEC), and BBM models [106]. To account for the distinct distribution patterns of the yews, climatic and geographical barriers, and flora zoning within the study area [48,107], we further refined the division of the distribution area into seven biogeographic regions based on the continental classification by Moller et al. [26]: (A) Northeastern China and Japan, (B) Eastern and Southern China, (C) Central China, (D) Hengduan Mountains, (E) Yunnan–Guizhou–Guangxi region and the Indo-China Peninsula, (F) Taiwan and the Philippine Islands, and (G) the Qinghai–Tibetan Plateau and Hindu Kush mountain regions (Figure A3b). The best-fit ancestral range model was identified using the BioGeoBEARS v2.1.1 method [108], based on the Corrected Akaike Information Criterion (AICc_wt).

### 4.6. Ancestral State Estimation

Ancestral states were estimated for 2 PCA-env components and 16 environmental variables to assess PNC. Phylogenetic signals (PSs) were evaluated using Blomberg’s K (K) and Pagel’s lambda (*λ*) with the R package “phytools v2.0” [109]. A K value greater than 1 indicates a strong PS with variance distributed between clades, while a K value less than 1 suggests a weaker PS with variance mainly within clades [110]. Pagel’s *λ* ranges from 0 (indicating no correlation) to 1 (indicating strong correlation) [111]. We tested three evolutionary models using the “geiger v2.0” R package [112], including BM, Ornstein–Uhlenbeck (OU), and Evolutionary Burst (EB), with the best-fit model selected using AICc criteria [113]. BM assumes trait correlation proportional to shared ancestry, OU implies stabilizing selection, and EB represents rapid early trait evolution followed by stasis. Models with ΔAIC < 2 were considered roughly equivalent, while ΔAIC values within 4–7 indicated distinguishable models and ΔAIC > 10 signified different models. Ancestral states were estimated using the “phytools v2.0” R package [109] under the best-fit model, focusing on seven climatic variables and three environmental factors with strong PS.

### 4.7. Predicted Niche Occupancy and Ancestral Tolerances

We utilized the “Phyloclim v0.9.5” R package [114] to assess the ancestral ecological tolerances and generate PNO profiles [115]. PNO profiles reflect species’ probability distributions from the eENM, indicating each lineage’s tolerance or occupancy concerning specific environmental variables. Through 1000 random iterations, we estimated ancestral tolerances to 10 key environmental variables.

A DTT analysis was performed to assess niche evolution dynamics and disparity among lineages, quantified using the MDI [116]. “Disparity” refers to the mean squared Euclidean distance of weighted mean PNO values between species pairs, while “relative disparity” normalizes clade-specific disparity against that of the entire phylogenetic tree. DTT was calculated as the mean relative disparity of all clades at each speciation event, considering ancestral lineages. An expected DTT distribution, based on a BM model, allowed us to compare observed and expected DTT values over divergence times in a DTT plot. The MDI measures the deviation of observed DTT values from those predicted. Positive MDI values indicate disparity concentrated within subclades or recent divergence, while negative values suggest a broader disparity between subclades or trait conservatism in deeper clades [115]. The DTT analysis was conducted using the “geiger v2.0” R package, with 1000 simulations to achieve a 95% confidence level [117].

## 5. Conclusions

This study highlights the need to integrate evolutionary and ecological approaches to understand the multidimensional traits and evolution of ecological niches, the biogeographical history, and the conservation priorities of rare relict species. The diversification pattern of East Asian *Taxus* is shaped by a combination of geographic, ecological, and genetic factors. Niche diversification within East Asian *Taxus* has been driven by in situ adaptation and dispersal from centers, such as Central China, the Hengduan Mountains, and the Yunnan–Guizhou Plateau, with orogenic and climatic changes in QTP during the Late Miocene playing a significant role. The key environmental factors influencing niche divergence include temperature, precipitation, light, and altitude. Both niche conservatism and divergence are evident within the genus, with early conservatism and recent divergence. The Southern clade shows significant niche divergence, while the Central and Western clades exhibit phylogenetic niche conservatism. Special attention is needed for *T. qinlingensis* due to its high niche conservatism and habitat disruption. We recommend strengthening fragmented protected areas to cover current gaps and regions vulnerable to future climate fluctuations, particularly the southern Himalayas and the Hengduan, Qinling–Daba, and Taihang Mountains. Future research should utilize advanced sequencing technologies and big data analytics to explore the biogeography and niche evolution of *Taxus* across multiple dimensions, providing a basis for optimizing conservation strategies.

## Figures and Tables

**Figure 1 plants-14-01094-f001:**
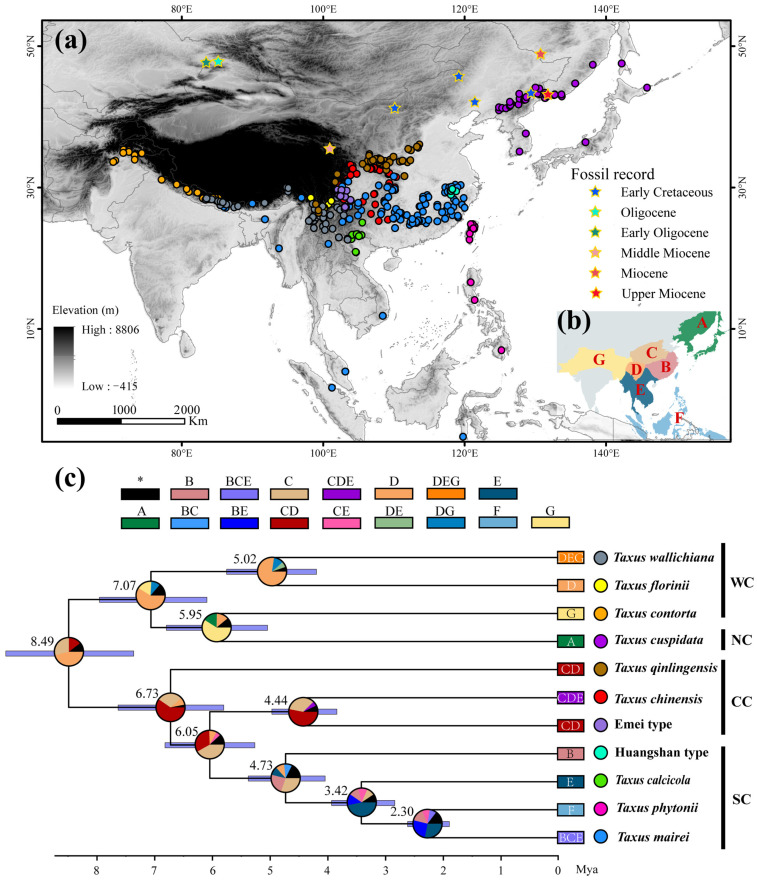
Geographic distribution, dated maximum clade credibility tree, and reconstructed ancestral range of East Asian *Taxus* lineages. (**a**) Geographic distribution of 11 *Taxus* lineages. Detailed information on all occurrence points and fossil records is provided in Appendix A. (**b**,**c**) Phylogenetic tree reconstructed using BEAST based on 13 chloroplast DNA (cpDNA) markers, ITS, and NEEDLY sequences, and ancestral range estimates inferred using a Bayesian Binary MCMC (BBM) model implemented in RASP v4.0. In (**c**), the boxes at the top and end of the phylogenetic tree correspond to the geographic areas shown in (**b**): (A) Northeastern China and Japan, (B) Eastern and Southern China, (C) Central China, (D) Hengduan Mountains, (E) the Yunnan–Guizhou–Guangxi region and Indo-China Peninsula, (F) Taiwan and the Philippine Islands, and (G) the Qinghai–Tibetan plateau and Hindu Kush mountains regions. Numbers at each node represent mean age estimates, with blue bars showing the 95% highest posterior density (HPD) confidence intervals. Corresponding divergence times and evolutionary events are detailed in Table A1. Pie charts indicate the estimated probability of each area being the ancestral range. Dots next to the phylogenetic tree represent lineages-specific occurrence points. WC, NC, CC, and SC represent the Western, Northern, Central, and Southern clades, respectively.

**Figure 2 plants-14-01094-f002:**
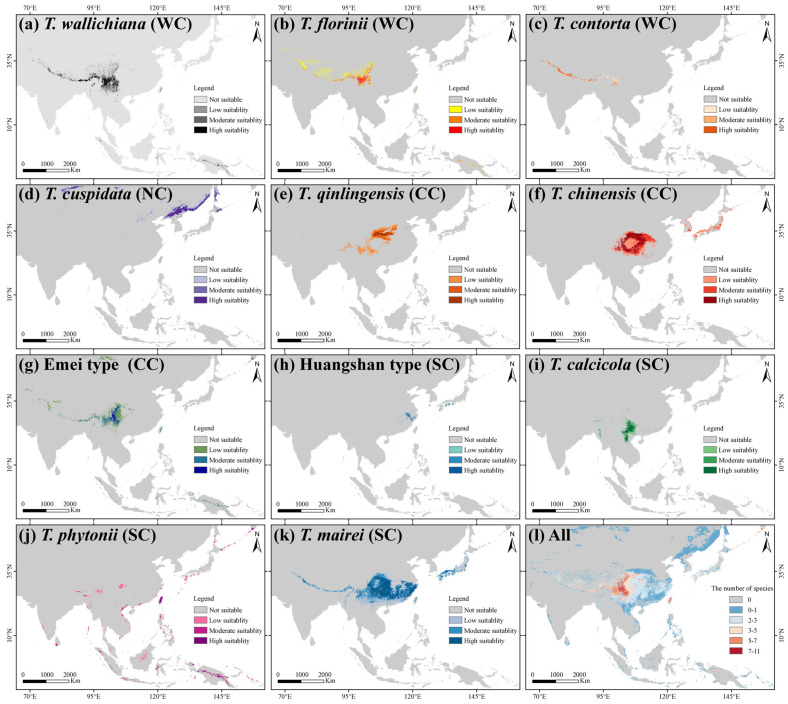
Geographic distribution of suitable habitats for 11 *Taxus* lineages in East Asia, as predicted by the biomod2 ensemble ecological niche model (eENM). The potential distributions were modeled using 16 environmental variables. WC, NC, CC, and SC represent the Western, Northern, Central, and Southern clades, respectively.

**Figure 3 plants-14-01094-f003:**
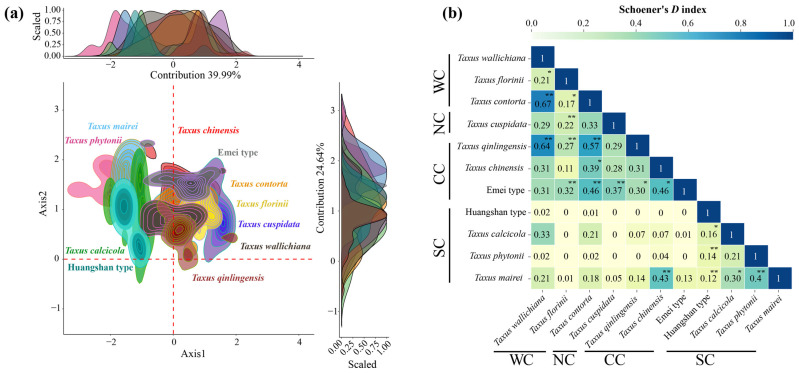
Niches of 11 *Taxus* lineages in East Asia in relation to environmental space. (**a**) The niche characteristics for the 11 *Taxus* lineages along the two first axes of the environmental principle components analysis (PCA-env); (**b**) summary of pair comparisons between observed values and 1000 randomly generated empirical values based on Schoener’s *D* index (significant values for niche identity and background similarity tests depicted as follows: * = significant in only one test and ** = significant in both tests [*p* < 0.05]; Figure A4 provides detailed results for pair comparisons among lineages and the two tests). WC, NC, CC, and SC represent the Western, Northern, Central, and Southern clades, respectively.

**Figure 4 plants-14-01094-f004:**
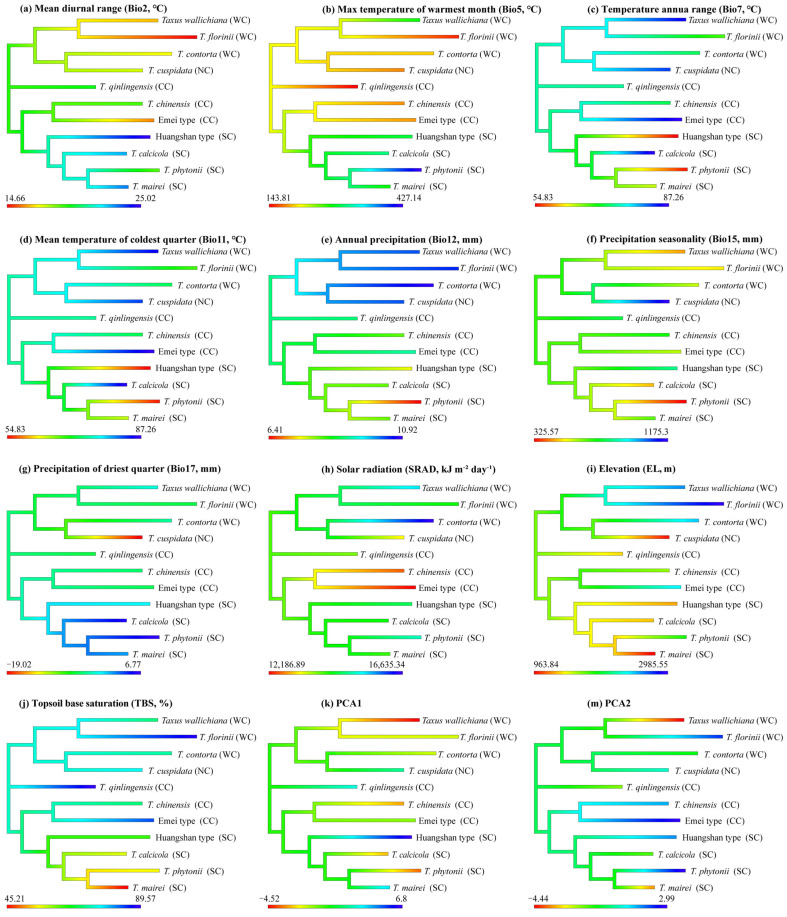
Niche ancestry reconstruction for 11 *Taxus* lineages in East Asia differing in 10 environmental variables and 2 PCs with a higher phylogenetic signal. Higher values are displayed in blue, intermediate values in green, and low values in red. WC, NC, CC, and SC represent the Western, Northern, Central, and Southern clades, respectively.

**Figure 5 plants-14-01094-f005:**
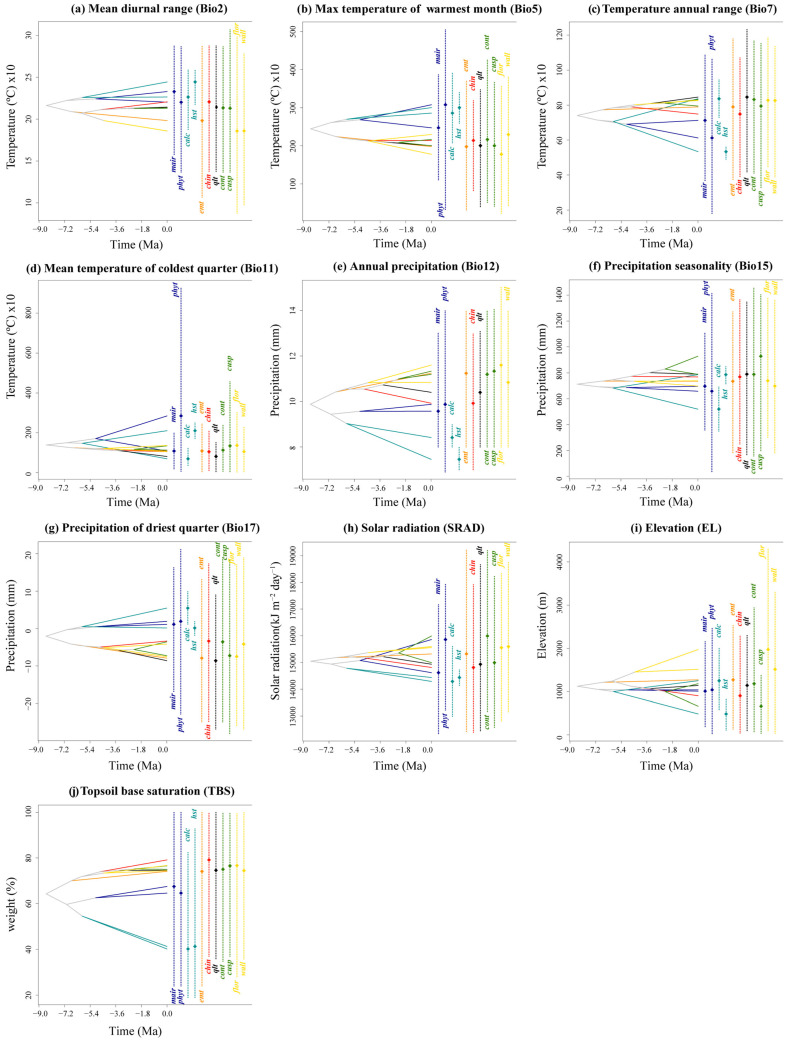
Inferred history of the evolution of environmental tolerances of ten environmental variables within the 11 *Taxus* lineages in East Asia based on a BEAST chronogram. Crossing branches of the phylogenetic tree indicate convergent niche evolution among taxa from different clades, and overlapping internal nodes indicate convergent environmental origins. Vertical dashed line and point marks show the 80% central density of environmental tolerance for each extant taxon and the related mean values, respectively. Lineage abbreviations are as follows: *Taxus wallichiana* (wall), *T. florinii* (flor), *T. cuspidata* (cusp), *T. contorta* (cont), *T. chinensis* (chin), *T. calcicola* (calc), *T. phytonii* (phyt), *T. mairei* (mair), *T. qinlingensis* (qlt), Huangshan type (hst), and Emei type (emt).

**Figure 6 plants-14-01094-f006:**
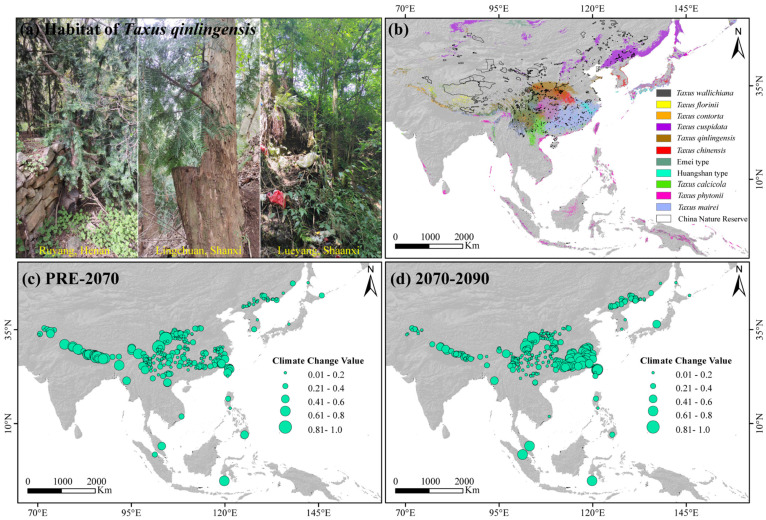
*Taxus qinlingensis* habitats (**a**), conservation gaps in 11 *Taxus* lineages in East Asia and nature reserves (**b**), and climate fluctuations for *Taxus* in East Asia from the present to 2090 (**c**,**d**).

**Table 1 plants-14-01094-t001:** Potential suitable areas and niche breadth of *Taxus* lineages in East Asia.

Clade	Taxon	Total Suitable Area	Low Suitability	Moderate Suitability	High Suitability	Niche Breadth
Western clade	*T.wallichiana*	3.47%	1.75%	0.83%	0.89%	0.57
	*T. florinii*	2.06%	1.25%	0.50%	0.32%	0.11
	*T. contorta*	0.69%	0.29%	0.15%	0.25%	0.23
Northern clade	*T. cuspidata*	4.65%	2.64%	0.92%	1.09%	0.48
Central clade	*T. qinlingensis*	4.64%	2.37%	1.11%	1.16%	0.42
	*T. chinensis*	5.67%	2.45%	1.55%	1.66%	0.70
	Emei type	3.37%	2.35%	0.77%	0.25%	0.05
Southern clade	Huangshan type	0.27%	0.08%	0.07%	0.12%	0.01
	*T. calcicola*	1.45%	0.53%	0.35%	0.56%	0.35
	*T. phytonii*	2.41%	1.44%	0.58%	0.40%	0.38
	*T. mairei*	11.05%	3.88%	3.24%	3.93%	0.79

**Table 2 plants-14-01094-t002:** Phylogenetic signals and alternative evolutionary model performance tests for 16 environmental variables.

	Blomberg’s K		Pagel’s Lambda (λ)				Brownian Motion			Ornstein–Uhlenbeck			Evolutionary Burst		
	K	*p*-Value	λ	logL	logL0	*p*-Value	−In *L*	AICc	ω	−In *L*	AICc	ω	−In *L*	AICc	ω
PC1	0.709	0.791	6.61 × 10^−5^	−29.139	−0.001	1	**−27.412**	**60.336**	**0.867**	−30.60	70.60	0.01	−27.40	64.30	0.12
PC2	0.644	0.908	6.61 × 10^−5^	−25.255	−0.001	1	**−22.210**	**49.819**	**0.867**	−27.46	64.35	0.48	−22.20	53.82	0.51
Bio2	** *1.221* **	** *0.002* **	** *0.971* **	−26.781	2.431	** *0.012* **	**−23.806**	**53.234**	**0.860**	−26.789	62.909	0.615	−23.760	56.915	0.137
Bio5	** *1.181* **	** *0.003* **	** *0.814* **	−63.182	1.904	** *0.006* **	**−58.678**	**122.875**	**0.867**	−63.245	135.971	0.011	−58.687	126.787	0.125
Bio7	0.810	0.407	6.601 × 10^−5^	−42.164	−5.991 × 10^−5^	1	**−40.314**	**86.158**	**0.807**	−43.101	95.641	0.700	−40.315	90.112	0.125
Bio11	** *1.035* **	** *0.009* **	** *0.970* **	−64.260	0.580	** *0.011* **	**−60.456**	**126.403**	**0.881**	−60.436	138.335	0.200	−60.462	130.345	0.125
Bio12	** *1.674* **	** *0.001* **	** *1.101* **	−15.857	5.466	** *0.002* **	**−15.931**	**37.321**	**0.766**	−15.895	41.113	0.122	−15.911	41.251	0.117
Bio15	0.815	0.525	6.61 × 10^−5^	−74.927	−9.162 × 10^−5^	1	**−69.295**	**114.000**	**0.886**	−79.925	161.362	0.157	−69.275	147.933	0.125
Bio17	** *1.084* **	** *0.004* **	** *0.985* **	−35.629	1.771	** *0.008* **	**−33.045**	**71.601**	**0.872**	−33.648	80.800	0.081	−33.045	75.529	0.125
SRAD	** *1.017* **	** *0.002* **	** *0.837* **	−93.216	0.007	** *0.001* **	**−86.110**	**177.700**	**0.881**	−86.105	180.213	0.000	−86.995	181.623	0.125
EL	** *0.980* **	** *0.007* **	** *0.896* **	−86.725	0.231	** *0.004* **	**−79.337**	**164.245**	**0.881**	−87.012	183.450	0.055	−79.347	168.137	0.137
SLO	0.813	0.713	6.601 × 10^−5^	−18.309	−4.281 × 10^−5^	1	**−17.612**	**40.717**	**0.850**	−18.987	47.402	0.013	−17.601	44.674	0.125
ASP	0.857	0.421	6.601 × 10^−5^	−57.273	−3.237 × 10^−5^	1	**−53.300**	**112.131**	**0.881**	−58.132	125.619	0.079	−53.301	116.053	0.125
NPP	0.784	0.662	6.601 × 10^−5^	−105.158	−9.006 × 10^−5^	1	**−96.201**	**197.190**	**0.887**	−106.150	222.462	0.014	−96.210	201.883	0.125
TBS	** *1.262* **	** *0.001* **	** *0.999* **	−42.895	3.670	** *0.006* **	**−40.734**	**86.968**	**0.863**	−42.829	95.221	0.123	−40.723	90.818	0.125
TRBD	0.977	0.168	0.625	19.776	0.758	0.382	16.475	−27.457	0.232	**19.595**	**−29.706**	**0.754**	16.477	−23.052	0.003
TCC	0.923	0.302	0.999	−9.111	1.615	0.207	−8.312	22.138	0.259	−9.111	27.647	0.012	**5.494**	**20.421**	**0.619**
TS	0.791	0.640	6.601 × 10^−5^	−26.800	−0.001	1	**−25.841**	**57.107**	**0.816**	−27.709	65.016	0.170	−25.874	61.150	0.122

Note: The means of Blomberg’s K (K) and Pagel’s lambda (λ) values are used to characterize the strength of the phylogenetic signals. Traits with values significantly different from zero are in both bold and italic. ln *L*: values for Neperian logarithm; AICc: corrected Akaike information criterion; ω: Akaike weights. The preferred models are in bold. Bio2, mean diurnal range; Bio5, maximum temperature of warmest month; Bio7, temperature annual range; Bio11, mean temperature of coldest quarter; Bio12, annual precipitation; Bio15, precipitation seasonality; Bio17, precipitation of driest quarter; SRAD, solar radiation; EL, elevation; SLO, slope; ASP, aspect; NPP, net primary productivity; TBS, topsoil base saturation; TRBD, topsoil reference bulk density; TCC, topsoil calcium carbonate; TS, topsoil cation exchange capacity.

**Table 3 plants-14-01094-t003:** Environmental variables used in ensemble ecological niche models.

Type	Code	Determinant Environmental Variables	Time Period	Source
Climatic variables	Bio2	Mean diurnal range (°C)	1991–2000	https://www.worldclim.org/ (accessed on 3 September 2023)
	Bio5	Max temperature of warmest month (°C)		
	Bio7	Temperature annual range (°C)		
	Bio11	Mean temperature of coldest quarter (°C)		
	Bio12	Annual precipitation (mm)		
	Bio15	Precipitation seasonality		
	Bio17	Precipitation of driest quarter (mm)		
Solar radiation variables	SRAD	Solar radiation (kJ m^−2^ day^−1^)		
Topographical variables	EL	Elevation (m)	2019	https://www.worldclim.org/ (accessed on 3 September 2023)
	SLO	Slope		
	ASP	Aspect (°)		
Biological interaction	NPP	Net primary productivity	2000–2016	https://neo.gsfc.nasa.gov/ (accessed on 3 September 2023)
Soil condition	TBS	Topsoil base saturation (% weight)	2009	http://www.fao.org/ (accessed on 3 September 2023)
	TRBD	Topsoil reference bulk density (kg/dm^3^)		
	TCC	Topsoil calcium carbonate (% weight)		
	TS	Topsoil cation exchange capacity (soil; cmol/kg)		

## Data Availability

The original distribution datasets in the current study are available in Appendix A. Sequence data downloaded from NCBI for phylogenetic analysis are shown in Table A4.

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
