# Peer review of "Multiple Ecological Niche Modeling Reveals Niche Conservatism and Divergence in East Asian Yew (*Taxus*)"

_plants, 2025, doi:10.3390/plants14071094_

Round 1
Reviewer 1 Report
Comments and Suggestions for Authors
Manuscript ID: plants-3470149
Title: Multiple ecological niche modeling reveals niche conservatism and divergence in Asian yew (Taxus): implications for lineage diversification and conservation priorities
General Remarks:
This study is foccussed on Asian jew species. The initial hypothesis is that the proposed methodology (ecological niche modelling and cpDNA analysis) is able to help with the conservation activities of these species. The integration of evolutionary and ecological approaches is interesting and a promising pair in future conservation programmes.However, many details of the methodology are not well explained or are inadequate. For example, the analysis of cpDNA does not specify how many samples were analyzed from each lineage and their representativeness of the selected species. In the ecological niche model, a grid of 2.5' is used, which is clearly insufficient for species with such limited distribution and relict characteristics. It may be adequate for widely distributed species like Pinus sylvestris, but not for the group of species addressed in this study. At least a resolution of 100 m for the dependent variable is needed to face the study for the selected species.
The results obtained, although somewhat coherent, can only be regarded as mere approximations of potential distribution. Furthermore, the number of presence records ranges from 5 to 54, which is also insufficient for an ecological niche modelling study. This analysis also includes fossil records to increase presence and contribute to lineage studies; however, the variables used pertain to the period 1991–2000 (Table 3). Were the climatic conditions of 1991–2000 applicable to those fossil records in order to obtain reliable data? In conclusion, I believe that the results obtained do not possess sufficient quality to be considered reliable. For instance, in the conclusions it is stated that orogenic an climatic changes during the late Miocene played a relevant role in the adaptation and dispersion of the species, but at the same time, temperature, precipitation (in the period 1991–2000) are the key environmental factors influencing niche divergence. It seems that you are carriying two different approaches that are difficult to confront with the proposed methodology. An alternative is to separate the study in two different studies:
- ENM of the Asian species of Jew with higher resolution (at least for the dependent variable) and projection in the frame of climate change (different scenarios using worldclim).
- clDNA analysis and phylogenetic study. In this study could be introduced the ENM techniques by hindcasting with different time slices proposed in worldclim (Last Interglacial (~ 130 ka BP), Last Glacial Maximum (~ 21 ka BP), mid-Holocene (~6 ka BP)). See Vesella et al., 2015, Quaternary Science Reviews. Volume 119, Pages 85-93, for further details.
Specific comments:
Abstract:
There is a lack of information about applied methodology.
Introduction:
Lines 51-61: This long explanation about the speciation process is unnecessary, this is not a dissertation focused on speciation that needs such details.
Line 67: It is refers that mountains are hotspots. It is figurative because hotspots has a clear definition in the conservation science. It is preferable to state: “Mountains could be considered as biodiversity hotspots, since they host…..”
Line 69: it is stated that rapid orogenic are the responsible of many species isolation, but glaciations, faster that orogenic, are not mentioned along the text and it sure that are more responsible of species isolation than orogenic for the jew studied species.
Lines 91-93. There are some populations of the European jew (Taxus baccata) in West Asia, please review.
Results:
Lines 237-238: It seems that BMM is misspelling by BBM (Bayesian Binary MCMC analysis).
Material and methods:
Lines 587-591: Genetic analysis is not well explained and justified. How many samples were analized? Are they representative of all the studied species?
Line 596: it is stated that only one record by 2.5’ by 2.5’. It means that is the spatial resolution of the analysis, not enough for relict/endemic species as these studied here.
Lines 608-613: the selection of time slices, scenarios and climatic model are not well explained and justified. Why have you selected this periods, scenarios, models, etc. and not others?
Author Response
The reply letter is in the attachment below.

Reviewer 2 Report
Comments and Suggestions for Authors
Please look at the attached document.

Author Response

(The authors gave the same response as above.)

Reviewer 3 Report
Comments and Suggestions for Authors
Dear Authors,
I read the paper of Wang and co-authors entitled Multiple ecological niche modeling reveals niche conservatism and divergence in Asian yew (Taxus): implications for lineage diversification and conservation priorities and I think that it is a solid paper which describe the divergence and ecological niche evolution of 11 Taxus species. I present below a few important points/issues to discuss by Authors.
- The title can be shorter and more complicated: Multiple ecological niche modeling reveals niche conservatism and divergence in Asian yew (Taxus).
- The Introduction section lacks the reason why the Authors would like to study the 11 Taxus lineages in geographic and environmental space. Why did they choose these taxa for analysis? I can see that in 85-90 lines, the Authors described this issue, but I think the conservation status of these taxa needs to be shown.
- I wonder why there are not so much distribution points for all taxa. What is the geographical distribution of 11 Taxus taxa in China? Is it really similar to the potential geographical distribution?
- The description for all figures needs to be completed. It is very hard to understand when no code descriptions exist on these pictures.
- Please list keywords in alphabetical order.
- Authors showed in Fig 1 (c) the phylogenetic tree without outgroup and estimated divergence time. At this point, this time of divergence can be very speculative. The good choice of outgroup species can give more probabilistic results, and the calibration point will be better validated. I hope that my suggestions help to improve this paper.
Yours sincerely, Reviewer
Author Response

(The authors gave the same response as above.)
